# Semantic Segmentation Guided Coarse-to-Fine Detection of Individual Trees from MLS Point Clouds Based on Treetop Points Extraction and Radius Expansion

**Xiaojuan Ning** [1,2,*], **Yishu Ma** [1], **Yuanyuan Hou** [1], **Zhiyong Lv** [1,2], **Haiyan Jin** [1,2] **and Yinghui Wang** [3]

1 Institute of Computer Science and Engineering, Xi'an University of Technology, No.5 South of Jinhua Road, Xi'an 710048, China
2 Shaanxi Key Laboratory of Network Computing and Security Technology, Xi'an 710048, China
3 School of Artificial Intelligence and Computer Science, Jiangnan University, 1800 of Lihu Road, Wuxi 214122, China
* Correspondence: ningxiaojuan@xaut.edu.cn

**Abstract:** Urban trees are vital elements of outdoor scenes via mobile laser scanning (MLS), accurate individual trees detection from disordered, discrete, and high-density MLS is an important basis for the subsequent analysis of city management and planning. However, trees cannot be easily extracted because of the occlusion with other objects in urban scenes. In this work, we propose a coarse-to-fine individual trees detection method from MLS point cloud data (PCD) based on treetop points extraction and radius expansion. Firstly, an improved semantic segmentation deep network based on PointNet is applied to segment tree points from the scanned urban scene, which combining spatial features and dimensional features. Next, through calculating the local maximum, the candidate treetop points are located. In addition, the optimized treetop points are extracted after the tree point projection plane was filtered to locate the candidate treetop points, and a distance rule is used to eliminate the pseudo treetop points then the optimized treetop points are obtained. Finally, after the initial clustering of treetop points and vertical layering of tree points, a top-down layer-by-layer segmentation based on radius expansion to realize the complete individual extraction of trees. The effectiveness of the proposed method is tested and evaluated on five street scenes in point clouds from Oakland outdoor MLS dataset. Furthermore, the proposed method is compared with two existing individual trees segmentation methods. Overall, the precision, recall, and F-score of instance segmentation are 98.33%, 98.33%, and 98.33%, respectively. The results indicate that our method can extract individual trees effectively and robustly in different complex environments.

**Keywords:** mobile laser scanning; individual trees extraction; semantic segmentation; MLS point clouds

## 1. Introduction

With the rapid development of MLS, the PCD obtained by MLS is widely used to express the 3D surface information of roadside objects [1,2]. The result of extracting trees individually and capturing the attributes of trees from MLS point clouds can be widely used in various applications, such as urban road planning, street tree 3D modeling [3], street tree monitoring [4], tree species identifying [5], and biomass estimation [6]. However, MLS point clouds of an outdoor scene are usually characterized by complex, diverse outdoor scene objects, and the different densities distribution of tree point cloud. Furthermore, trees usually are spatially overlapping with other non-tree objects (e.g., lamps, billboards) and tree crowns. These attributes pose significant challenges to detect individual trees from scanned outdoor scene.

In recent years, some automated methods based on MLS have been proposed [7]. There are many scientific contributions aiming to segment scanned urban scenes into different objects [8–10] and capture the attributes of trees [11–15] (e.g., tree height, trunk diameter

and diameter at breast height), and more outstanding work on 3-D object detection based on LiDAR data emerges [16]. In this work, we focus on the current methods for individual trees detection from MLS. These methods can be roughly divided into three categories, i.e., the normalized cut methods (NCut) [17–20], the region growing methods [21–24], the clustering-based methods [25–28].

To improve the classification accuracy, Xu et al. [18] spatially smoothed the semantic label results obtained by Random Forest classifier via a regularization process, and then extracted the individual trees based on NCut. However, NCut only considers the distance, resulting in inaccurate boundary segmentation of tree crowns. In addition, the prior knowledge of the number of trees cannot be obtained, over-segmentation and under-segmentation are prone to occur. Zhong et al. [19] used the improved NCut to segment overlapping regions to obtain the individual trees. However, when there are poles near the trunk, the height threshold has a great influence on the trunk detection. The individual trees detection based on NCut needs to manually estimate the number of trees in a multi-tree cluster to determine the iteration termination condition. NCut requires large storage space and is inefficient when the PCD is dense, so NCut is mostly used for fine segmentation of under-segmentation overlapping objects.

Bonneau et al. [23] divided the PCD into voxels and clustered the connected voxel units based on region growing. And then judged whether this was correctly segmented by analyzing the spatial range and eigenvalue ratio of the clustering units, to further refine the under-segmented clusters and merge the over-segmented clusters. However, this method requires complete tree structure information, and it will fail when the tree data is incomplete. Luo et al. [24] proposed a deep network for semantic segmentation to extract tree points from raw point clouds. A pointwise direction embedding deep network (PDE-Net) is proposed to predict the direction vector of each tree cluster pointing to the tree center to improve the tree boundary segmentation accuracy. On this basis, a tree center detection method based on pointwise direction aggregation is proposed, and finally, extract individual trees based on the detected tree center as the seeds of the region growing. However, the direction prediction is inaccurate when the classification accuracy of tree points is low, and satisfactory extraction results cannot be obtained. A region growing-based method may not be able to obtain correct segmentation results due to improper selection of seeds or inaccurate feature extraction. Especially when trees are adjacent to some pole-like objects, it is difficult to separate trees and roadside pole-like objects, so there are major flaws in extracting individual trees from complex outdoor scenes.

Yang et al. [25] extracted the treetop points by 3-D spatial distribution analysis and used the treetop points as the seeds of the k-means clustering to segment the individual trees. However, k-means clustering requires the number of clustering as an input parameter. Tao et al. [26] intercepted PCD of a certain height and used DBSCAN clustering to obtain tree trunks. However, the trunks extraction result is unsatisfactory when the data density is uneven. Chen et al. [28] extracted individual trees based on the Euclidean clustering. The Euclidean clustering does not require prior knowledge of the number of trees in the clustering process, the Euclidean distance of adjacent points needs to be compared with a user-defined threshold, which is difficult to set. When the threshold is small, tree points may be lost or over-segmented into multi-tree clusters. On the contrary, objects close to the tree cannot be separated. It is easy to cluster multiple connected trees together in complex outdoor scenes and produce over-segmentation when the tree data is missing. In addition, the clustering based methods also have certain limitations. For example, k-means based tree extraction requires the number of trees and the initial clustering position in advance. When the data is missing or the parameters are set incorrectly, the segmentation of the DBSCAN will be affected. Therefore, prior knowledge and parameter settings are very important factors when using the tree extraction method based on clustering.

Ncut has high time complexity when it dealing with complex scenes. Compared with NCut, region growth makes full use of the local features of point clouds for segmentation, but the segmentation effect depends on the growth criteria and seeds selection,



and it is difficult to segment correctly when trees and pole-like objects are close to each other. Clustering-based methods can achieve better extraction results in simple scenes, but under-segmentation occurs when trees are densely distributed, and over-segmentation occurs when point cloud data is incomplete. In conclusion, for outdoor scenes with large tree spacing and small overlap between tree crowns and nearby objects, most existing methods can segment and extract individual trees well. However, in complex scenes where multiple trees are connected or trees are adjacent to other objects, the extraction results of an individual tree is unsatisfactory. In addition, the current method is also affected by the density of point cloud data, which affects the extraction results of individual trees when the point cloud data is missing or incomplete.

In this paper, to overcome the problem of low tree extraction accuracy caused by the uneven density, missing or incomplete of point clouds, we proposed a novel method which combined tree detection with multi-feature enhanced PointNet, treetop points detection and radius expansion, to achieve a coarse-to-fine individual trees extraction from MLS point clouds. The main contributions of the proposed method are as follows.

(1) A comprehensive framework combining semantic segmentation, treetop points locating, and radius expansion is constructed for individual trees extraction. It can accurately extract an individual tree and solve the over segmentation caused by incomplete point cloud data and uneven density.

(2) A tree detection method based on the semantic segmentation by multi-feature enhancement PointNet is proposed to solve the classification of multiple-categories objects in complex outdoor scenes.

(3) A novel individual trees extraction method is introduced for scanned urban scene. Through calculating the local maximum, the candidate treetop points are extracted. Taking the treetop points as center, the radius expansion guided method is presented for further extraction of an individual tree.

## 2. Materials and Methods

The proposed method mainly contains three steps: (1) tree detection based on the semantic segmentation by multi-feature enhancement PointNet, (2) optimal treetop points location based on projection, and (3) individual trees detection based on radius expansion. The overview of our proposed framework is shown in Figure 1.

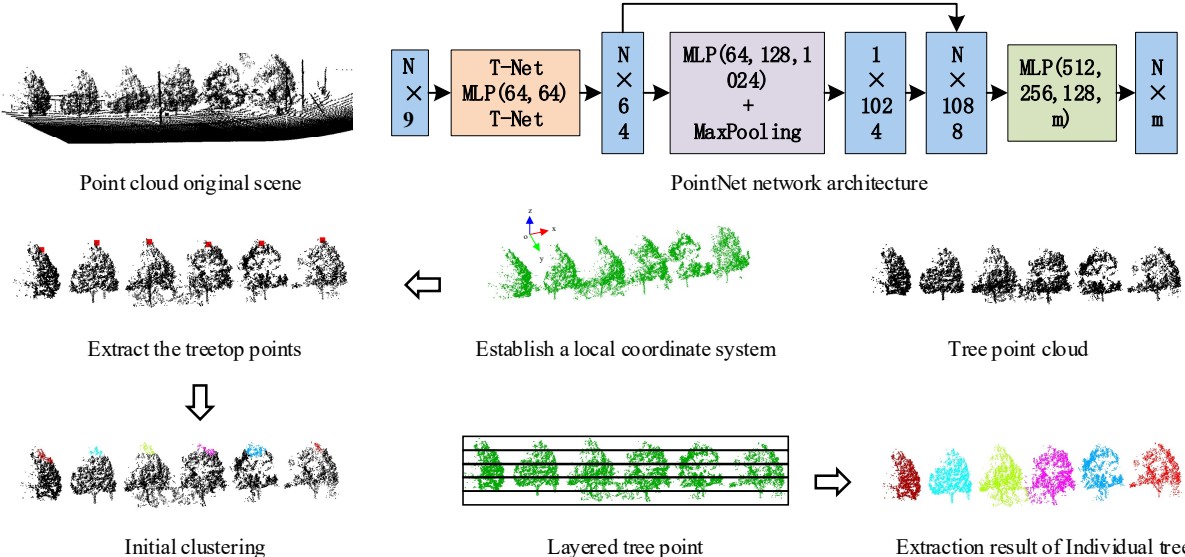

**Figure 1.** Overview of our proposed method.

### 2.1. Tree Points Detection Based on Multi-Feature Enhanced PointNet Semantic Segmentation

Generally, there are various objects in the outdoor scene, such as trees, buildings, ground, poles, vehicles, and etc. Therefore, it is necessary to remove non-tree objects in the scene and extract the tree points before extracting an individual tree. With the development of deep learning, Qi et al. [29] proposed a PointNet network that can directly process point cloud, which showed high accuracy and efficiency in semantic segmentation. We could detect trees from raw outdoor scene point cloud based on PointNet deep neural network. However, it only uses the Multilayer Perceptron (MLP) to increase the feature dimension when the PointNet model extracts local features and does not consider the neighborhood information of the point cloud, resulting in a poor description of the extracted local features. Therefore, the local features of the PCD are extracted and the coordinate values are combined to form feature vectors as the input of the PointNet network to perform semantic segmentation of complex outdoor point cloud scenes.

#### 2.1.1. 3D Point Cloud Features Extraction

The performance of 3D PCD local features description depends on its local neighborhood information. At present, the selection of point cloud neighborhood data can be roughly divided into two methods, i.e., the k-nearest neighbor (KNN) search algorithm and the spherical local search algorithm. KNN method is a density-adaptive search algorithm that takes the k points closest to the query point as neighborhood points and can obtain a consistent number of neighborhood points in the case of uneven point cloud density, which is beneficial to improve data storage and calculation efficiency.

Given scanned scene data $P = \{p_i | i = 1, 2, \ldots, N\}$, the k neighboring points of a point $p_i$ be $q_j = \{(x_j, y_j, z_j) | j = 1, 2, \ldots, k\}$. The normal vector estimation is implemented by a least-square plane fitting on the nearest neighbors, which is mainly based on Principal Components Analysis (PCA). Therefore, the local covariance matrix M of $p_i$ is constructed as:

$$M = \frac{1}{N} \sum_{i=1}^{N} (p_i - \overline{P})(p_i - \overline{P})^T \tag{1}$$

where $N$ is the number of points in the point cloud, $\overline{P}$ is the centroid point of the PCD, which is calculated by $\overline{P} = 1/N \sum_{i=1}^{N} p_i$. The eigenvalues are positive and ordered as $\lambda_1 \geq \lambda_2 \geq \lambda_3 \geq 0$. The normal vector $(n_i^x, n_i^y, n_i^z)$ of point $p_i$ can be determined by the eigenvector corresponding to $\lambda_3$. Ning et al. [30] applied the local features calculated by the covariance matrix to the machine learning classification algorithm for tree extraction and achieved good classification results. Based on this, we selected 6 features that have a strong description ability for outdoor scene PCD, namely linearity $L_\lambda$, flatness $F_\lambda$, divergence $D_\lambda$, anisotropy $A_\lambda$, characteristic entropy $E_\lambda$, and curvature variation $C_\lambda$ [31], these features can be calculated by Equation (2):

$$\begin{cases} L_\lambda = \left(\sqrt{\lambda_1} - \sqrt{\lambda_2}\right) / \sqrt{\lambda_1} \\ F_\lambda = \left(\sqrt{\lambda_2} - \sqrt{\lambda_3}\right) / \sqrt{\lambda_1} \\ D_i = \sqrt{\lambda_3} / \sqrt{\lambda_1} \\ A_i = (\lambda_1 - \lambda_3) / \lambda_1 \\ E_\lambda = -\sum_{i=1}^{3} \lambda_i \ln \lambda_i \\ C_i = \lambda_3 / (\lambda_1 + \lambda_2 + \lambda_3) \end{cases} \tag{2}$$

As we all know, the divergence, characteristic entropy, and curvature variation of trees are significantly higher than those of ground and buildings, while the linearity, flatness, and anisotropy of trees are lower than those of poles, buildings, and other objects. Therefore, the characteristics of different objects can be grasped more comprehensively and effectively through multi-feature fusion, and the discrimination between different objects can be improved.

### 2.1.2. PointNet Enhanced by Multi-Features

The disorder of the PCD makes the point cloud of different input orders get different high-dimensional features after passing through the MLP layer, which affects the feature extraction of the deep neural network. The rigid body invariance of the point cloud makes the spatial structure and shape information of the point cloud unaffected under different perspectives. Therefore, Qi et al. [29] introduced a T-Net module and a symmetric function to reduce the influence of disordered point clouds on the segmentation results. The specific steps of semantic segmentation are as follows. The PointNet network architecture diagram is shown in Figure 1.

The spatial coordinate features of N points are combined with local features in the data preparation stage and the input data is represented by a 9-D vector $\{X, Y, Z, L_\lambda, F_\lambda, D_\lambda, A_\lambda, E_\lambda, C_\lambda\}$. To adapt to the new number of channels, change the T-Net (3) of the PointNet network to T-Net (9), and then multiply the original PCD by the $9 \times 9$ transformation matrix learned by T-Net (9) to get the aligned data. After data alignment, the information of each point is learned and extracted by the MLP layer shared by 2 layers, and an $N \times 64$ matrix is obtained. Finally, the $64 \times 64$ feature space transformation matrix is predicted by T-Net (64), and the transformation matrix is applied to the $N \times 64$ matrix to achieve feature alignment, and the aligned features are used as the local features of the point cloud.

Input an $N \times 64$ matrix in the shared MLP, and map the data to 64-D, 128-D, and 1024-D in turn to obtain an $N \times 1024$ matrix. Then, the maximum value of N data in each dimension is extracted through the max pooling operation to obtain the global features of the point cloud. The aligned $N \times 64$ local features and $1 \times 1024$ global features are spliced through the fully connected layer to obtain an $N \times 1088$ matrix. Then, the three-layer MLP is used to classify and output the data, and a matrix of $N \times m$ is obtained, where N is the number of point clouds, m is the number of categories, and finally the semantic segmentation task of the scene is realized.

### 2.1.3. Filtering and Optimization

There are noisy points in the tree PCD obtained by semantic segmentation, so a filtering algorithm needs to be used to remove them. We use the pass-through filtering algorithm and the statistical filtering algorithm in Point Cloud Library (PCL) [32] to denoise the tree points. The statistical filter is mainly aimed at scattered noise points with a small amount of data. By calculating the average distance from each point to the adjacent points, and then comparing it with the given mean and variance, the noise points outside the range are eliminated. The pass-through filter can quickly remove a large number of outliers beyond the set range by determining the extent of the PCD on the X, Y, and Z axes. The comparison diagram of filtering is shown in Figure 2.

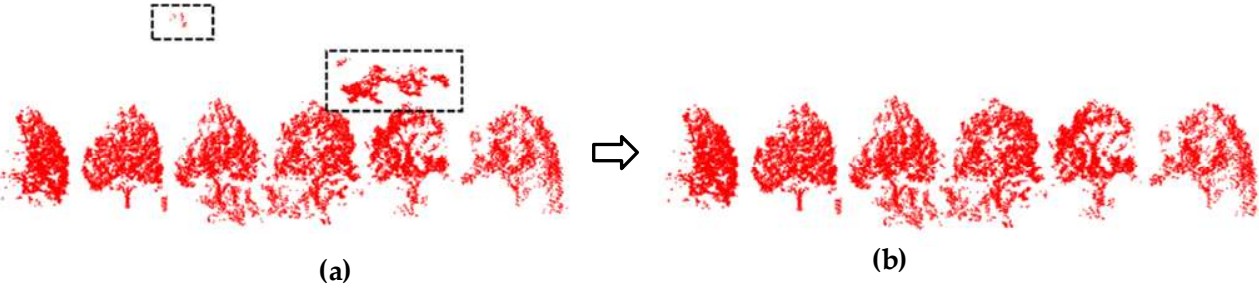

(a)　　　　　　　　　　　　　　　　　　　　　　　(b)

**Figure 2.** Trees points before and after filtering. (**a**) Trees points before filtering. (**b**) Trees points after filtering.

### 2.2. Treetop Points Extraction

Treetop points is the local highest points of crowns, which could determine the number of trees in the tree points from scene. Since there are often certain gaps between trees in urban scenes, even if the tree canopy overlaps. As the elevation increases, the horizontal spacing between the treetop points of different trees will become larger and larger. For single-row trees, it can be found that the treetop points of street trees are mostly located on the vertical plane of the tree distribution direction. According to the distribution of trees in outdoor scenes, we proposed a novel method to extract treetop points through local coordinate system (LCS) establishment, projection, and local maxima calculation.

#### 2.2.1. Projection Direction

The trees in the outdoor scene have the characteristics that the treetop points are always the highest points. To extract accurate treetops, it is necessary to project all the trees points especially for the single-row street trees. As we all known, the outline of the projected trees is approximate to an ellipse, and the treetops are mainly located on the long axis of the ellipse. Therefore, the LCS of trees can be constructed by PCA, i.e., $v_1$, $v_2$, and $v_3$ (corresponding to $\lambda_1 \geq \lambda_2 \geq \lambda_3 \geq 0$) represents the $x$-axis, $y$-axis, and $z$-axis, respectively. Then the plane where the $x$-axis and $z$-axis are located is selected as the projection direction.

Assume that the tree PCD in outdoor scene is $T = \{t_i | i = 1, 2, \ldots, N_t\}$. The centroid $\overline{T}$ of all the data in $T$ is calculated by $\overline{T} = \frac{1}{N_t}\left(\sum_{i=1}^{N_t} t_i\right) = \left(\overline{t}^x, \overline{t}^y, \overline{t}^z\right)$, where $N_t$ is the number of tree points and $t_i = \left(t_i^x, t_i^y, t_i^z\right)$, $t_i \in T$. Then, the tree points are projected onto the XOZ plane, and the point set after projection is $T' = \{t_i' | i = 1, 2, \ldots, N_{t'}\}$, which is shown in Figure 3.

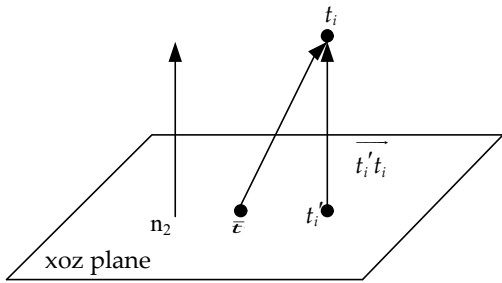

**Figure 3.** Tree points projection.

We can calculate the coordinate of $T' = \{t_i'(t_i'^x, t_i'^y, t_i'^z) | i = 1, 2, \ldots, N_{t'}\}$ by the Equation (3):

$$\begin{cases} t_i'^x = t_i^x - a \times \dfrac{l}{\sqrt{\|\vec{n_2}\|}} \\[2mm] t_i'^y = t_i^y - b \times \dfrac{l}{\sqrt{\|\vec{n_2}\|}} \\[2mm] t_i'^z = t_i^z - c \times \dfrac{l}{\sqrt{\|\vec{n_2}\|}} \end{cases} \tag{3}$$

where $\vec{n_2} = (a, b, c)$ is the normal vector of the XOZ plane. $l = \|\vec{t_i' t_i}\| = \left(t_i^x - \overline{t}^x\right) \times a + \left(t_i^y - \overline{t}^y\right) \times b + \left(t_i^z - \overline{t}^z\right) \times c / \sqrt{\|\vec{n_2}\|}$.

The LCS is established for the single row of outdoor trees data (shown in Figure 4a,4b). We projected the tree points onto the XOY plane (shown in Figure 4c) and the XOZ plane (shown in Figure 4d) of the LCS, respectively. We can see that it is easier to extract the treetops of the tree by projecting onto the XOZ plane.

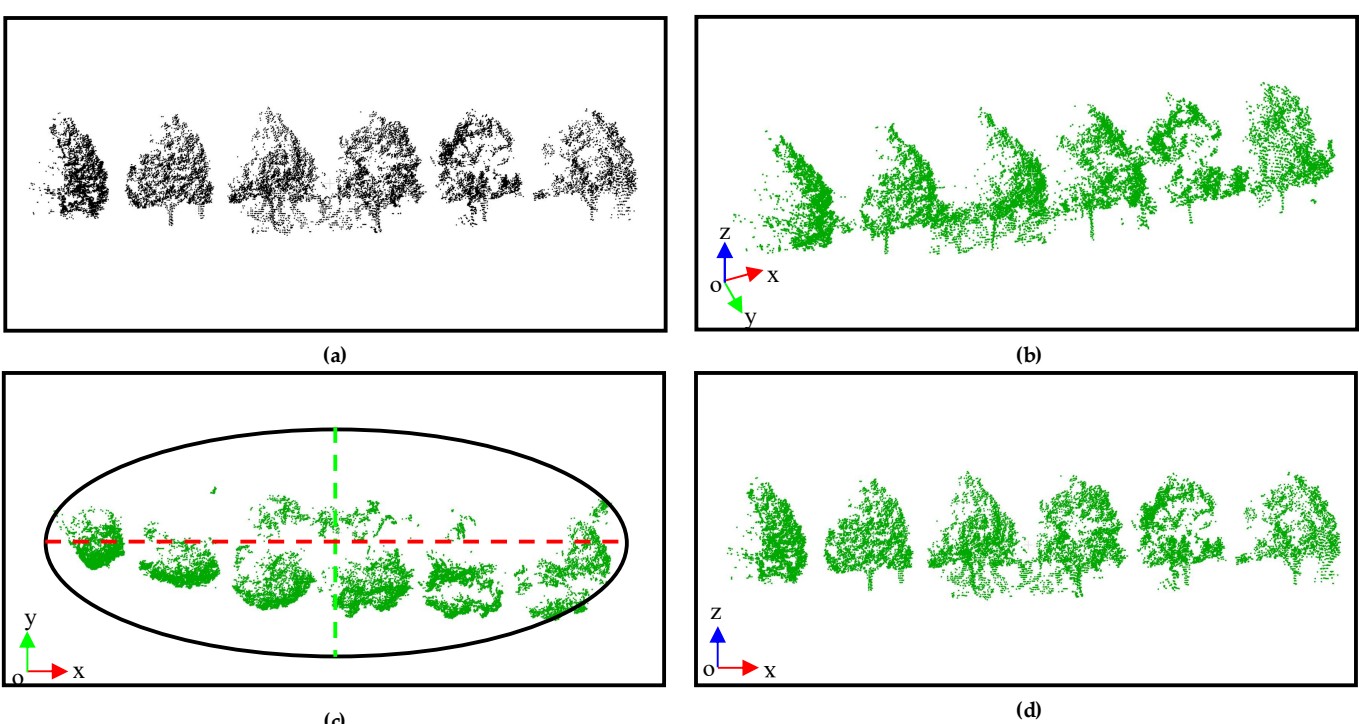

**Figure 4.** Projection of tree PCD. (**a**) is the point cloud of trees, (**b**) is the horizontal projection of tree clusters, (**c**) is the LCS establishment, (**d**) is the XOZ plane projection.

### 2.2.2. Optimal Treetop Points Extraction

The tree points data after projecting onto the XOZ plane could provide an easy way to obtain the most unobstructed treetop points and is a good representation of the shape of the tree canopy. Based on this, we propose an optimal treetop points extraction method by three steps: (1) local maxima calculation. (2) candidate treetop points locating. (3) optimal treetop points extraction.

(1)    Local maxima calculation

For the projection points on the XOZ plane, it is necessary to extract local maxima from the projection points to reduce the extraction range of treetop points and improve computational efficiency. Firstly, for points $t_i'\left(t_i'^x, t_i'^y, t_i'^z\right)$ and $t_j'\left(t_j'^x, t_j'^y, t_j'^z\right)$ on the projected contour, the redundant data are removed. That is to say, if $t_i'^x = t_j'^x$ and $t_i'^z = t_j'^z$, one of the points are kept. If $t_i'^x = t_j'^x$ and $t_i'^z < t_j'^z$, remove the point $t_i'$. Then sort all the projected points in ascending order of x coordinate to get the point set $TS' = \left\{ ts_i' \mid i = 1, 2, \ldots, N_{ts'} \right\}$. Next, $ts_i'$ is defined as the local maxima when $ts_i'^z > ts_{i-1}'^z$ and $ts_i'^z > ts_{i+1}'^z$, and the above procedure are repeated to extract all local maxima.

Figure 5 displays the comparison of local maxima extraction results before and after filtering redundant data. Figure 5a,b shows the raw PCD and the local maxima before filtering. Figure 5c indicates the data after the redundant points are removed, and the further extracted local maxima are shown in Figure 5d. It is worth noting that the local maxima obtained from the filtered scene are located on the outer contour position of the tree crown and more conducive to the extraction of subsequent treetop points.

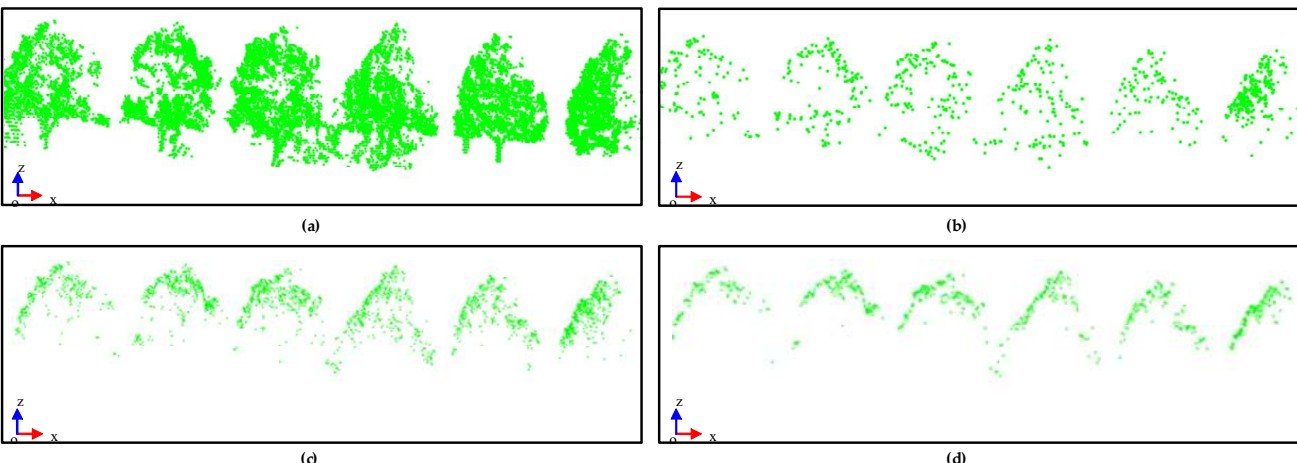

**Figure 5.** Comparison of local maximum extraction results. (**a**) PCD before filtering. (**b**) local maxima found before filtering. (**c**) PCD after filtering. (**d**) local maxima found after filtering.

(2)   Candidate treetop points locating

Based on the local maxima, a critical step in treetop points extraction is to locate candidate treetop points. For one tree, the changing trend of the crown contour points is to expand outward from the treetop points. Therefore, we locate the candidate treetop points according to the variation of the z-coordinate of the PCD. First, all local maximums are sorted in ascending order of x, denoted as $LM = \{m_i | i = 1, 2, \ldots, N_m\}$. $N_m$ is the number of the points of local maximum. Then, the difference $DM_i$ of point $m_i\left(m_i^x, m_i^y, m_i^z\right)$ on the z-axis are calculated by Equation (4).

$$DM_i = \begin{cases} m_{i+1}^z - m_i^z, i = 1 \\ \left(m_{i+1}^z - m_{i-1}^z\right)/2, i \in [2, 3, \ldots, N_m - 1] \\ m_i^z - m_{i-1}^z, i = N_m \end{cases} \tag{4}$$

Theoretically, the treetop point generally has maximum value of x coordinate among all its neighborhood points. Therefore, we locate candidate treetop points by detecting those points where their $DM$ varies from positive to negative. As the *x*-axis coordinate value continues to increase, there will be some randomly distributed noise points on the z-axis coordinate. To reduce the influence of noise, the difference needs to be smoothed. Thus, a two-step-based method is developed to detect candidate treetop points, i.e., the noisy points are deleted by smooth filtering (step 1), and then judge the symbol of DM. In step 1, for point $m_i$ we search its *k* nearest neighboring points and calculate the average difference $\overline{DM}$ and get the smoothed difference $DM'$. In step 2, the sign function is used to judge the positive and negative of smoothed difference of the point $m_i$:

$$sign(m_i) = \begin{cases} 1 & DM' > 0 \\ 0 & DM' = 0 \\ -1 & DM' < 0 \end{cases} \tag{5}$$

If $sign(m_i) > sign(m_{i+1})$, it means that the smooth difference will change from positive to negative, so the point $m_{i+1}$ is regarded as a candidate treetop point. This procedure is repeated, and then the set of candidate treetop points is obtained as $S = \{s_i | i = 1, 2, \ldots, N_s\}$ (see Figure 6a).



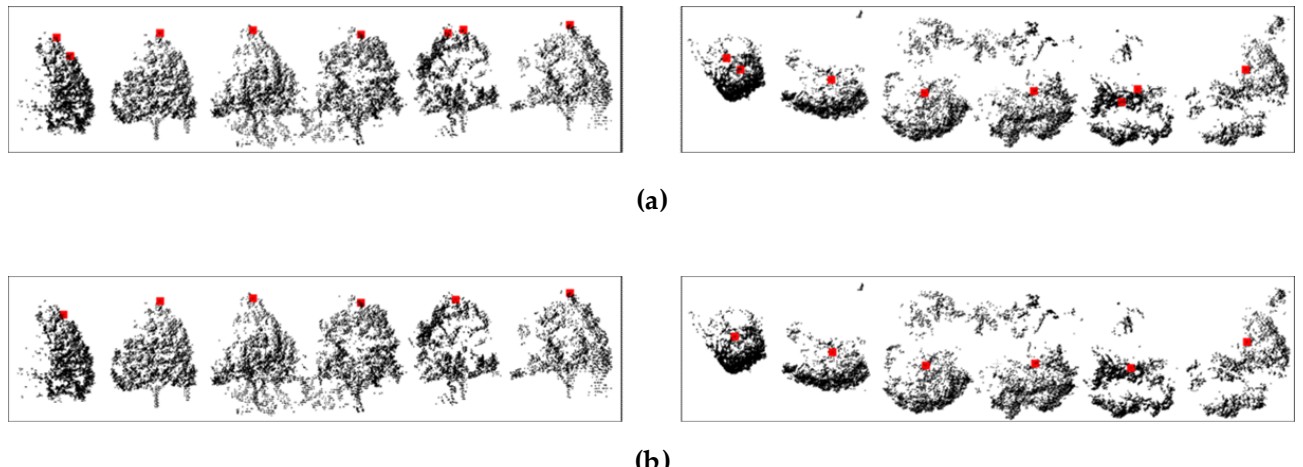

**Figure 6.** Comparison of treetop points optimization results. (**a**) Candidate treetop points extraction result. (**b**) Optimal treetop points extraction result.

(3)    Optimal treetop points extraction

The candidate treetop points obtained not only contain correct treetop points but also include some local extreme points with lower heights or redundant points with close distances between data. Thus, it is important to filter or remove those data that do not belong to the real treetop points. In our paper, two criterions are introduced. One is tree height and the other is distances between treetop points.

To begin with, the points that do not conform to the tree height are eliminated by judging whether the z-axis coordinate of each candidate treetop point is less than the height threshold $z_{th}$. We set the treetop point to be above $1/2$ of the height of the entire tree scene, and calculate the height threshold $z_{th}$ according to the Equation (6):

$$z_{th} = (z_{max} - z_{min})/2 + z_{min} \tag{6}$$

where $z_{max}$ and $z_{min}$ are the maximum z-coordinate and the minimum z-coordinate of tree PCD, respectively.

Then judged the distance between each pair of candidate treetop points. Those treetop points that are very close to each other are merged and optimized. Calculate the Euclidean distance between all candidate treetop points, and then sort the distances in ascending order. If the distance between the nearest pair of treetop points is less than the distance threshold $d_{th}$ (value is 0.5 m), the current two treetop points are replaced with their center points. After that, the distance between the updated treetop points is recalculated and evaluated. The optimized treetop points (see Figure 6b) are obtained until the distance between each two candidate treetop points are greater than $d_{th}$.

The front view and top view of the candidate treetop points extracted from Figure 4d are shown in Figure 6a. The optimal treetop points obtained after filtering and merging the candidate treetop points are shown in Figure 6b.

*2.3. Radius Expansion Based Individual Tree Extraction*

The challenge task of individual tree extraction is instance-level separation for spatially overlapping tree points [24]. After getting all the treetop points in the scene, we extract an individual tree in the outdoor scene based on the radius expansion.

The core steps of our proposed algorithm include initial clustering by analyzing the treetop points, initial bounding box and expansion circle determination, high-level layering for tree PCD and individual trees extraction by radius expansion.

Given the optimal treetop points $G = \{g_i | i = 1, 2, \ldots, N_g\}$, $N_g$ is the number of the treetop points. The initial clustering is carried out with treetop points as the center. The

specific steps are as follows: First, establish a KD-tree (k-dimensional tree, KdTree) with the point $g_i \in G$ as the seed point. Then take the seed point as the center of the sphere and set the radius of the ball to $IR(IR = 2m)$. Next, cluster the data points in the range of $IR$ with the seed points (Figure 7) to get the initial cluster $Clu_i$. In addition, this process is executed iteratively, until the spherical neighborhood data of all treetop points are divided, and the clusters are obtained, as shown in Figure 7.

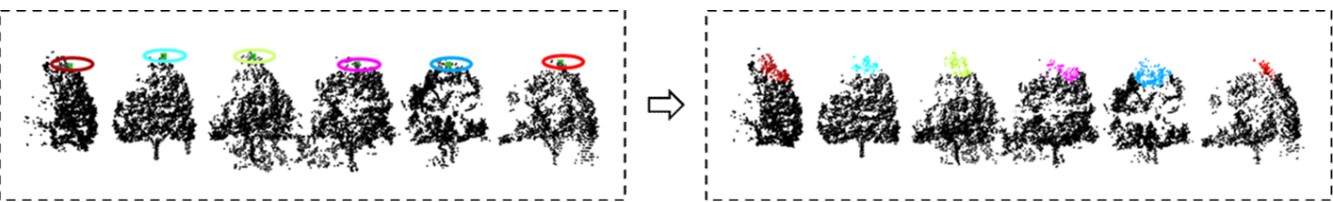

**Figure 7.** Initial clustering based on treetop points.

The purpose of clustering is to get the initial position where the bounding box and the extended circle are located. According to the initial clustering results, the maximum $(x_{max_i}, y_{max_i})$ and minimum $(x_{min_i}, y_{min_i})$ of all points in the cluster $Clu_i$ to form the initial boundary set $Bou_i$, $i \in [1, N_g]$. We calculate the radius $R_i$ and the center $O_i\left(O_i^x, O_i^y\right)$ of the extended circle where the cluster $Clu_i$ is located according to Equations (7) and (8), respectively.

$$R_i = \left(\left(x_{max_i} - x_{min_i}\right) + \left(y_{max_i} - y_{min_i}\right)\right)/4 \tag{7}$$

$$\begin{cases} O_i^x = \left(x_{max_i} - x_{min_i}\right)/2 \\ O_i^y = \left(y_{max_i} - y_{min_i}\right)/2 \end{cases} \tag{8}$$

After that, the boundary set of all clusters is $Bou = \{Bou_i \mid i = 1, 2, \ldots, N_g\}$, the radius of the expansion circle is $R = \{R_i \mid i = 1, 2, \ldots, N_s\}$ where the cluster is located, and the center of the circle is $O = \{O_i \mid i = 1, 2, \ldots, N_s\}$.

It is necessary to slice the tree PCD in the scene after obtaining the initial boundary, as shown in Figure 8. Set the number of split layers to $N_l$, then extract the maximum $z$-axis coordinate $z_{max}$ and the minimum $z$-axis coordinate $z_{min}$ from the tree points and calculate the height $H_l$ of each layer of data according to Equation (9).

$$H_l = (z_{max} - z_{min})/N_l \tag{9}$$

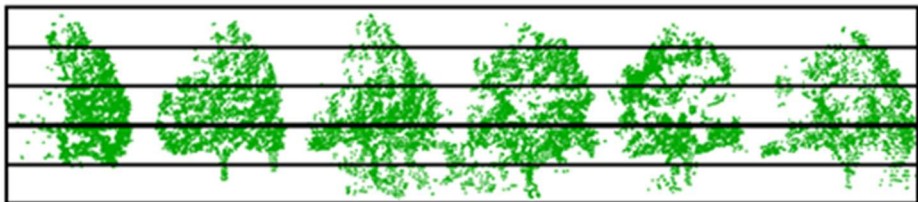

**Figure 8.** The schematic diagram of tree layering.

The point set of each layer from top to bottom is $L = \{L_i \mid i = 1, 2, \ldots, N_l\}$, the number of points in the $L_i$ layer is $N_{L_i}$. There are two cases to segment tree points:

(1) If point $t_u^{L_i}\left(t_{ux}^{L_i}, t_{uy}^{L_i}\right)$ in layer $L_i$ is within the range of $Bou_j$ of the cluster $Clu_j$ in layer $L_{i-1}$, and the horizontal distance $d^H\left(t_u^{L_i}, O_j\right)$ from point $t_u^{L_i}$ to the center $O_j\left(O_{jx}, O_{jy}\right)$

is less than the radius $R_j$, then point $t_u^{L_i}$ belongs to the tree where cluster $Clu_j$ is located, and point $t_u^{L_i}$ is assigned to cluster $Clu_j$, $d^H$ is calculated by Equation (10):

$$d^H\left(t_u^{L_i},O_j\right) = \sqrt[2]{\left(t_{ux}^{L_i}-O_{jx}\right)^2+\left(t_{ny}^{L_i}-O_{jy}\right)^2} \tag{10}$$

where $t_u^{L_i}$ is the $u$-th ($u \in [1,N_{L_i}]$) point of the layer $L_i$, $O_j$ ($O_j \in O$) is the center of the cluster $Clu_j$ ($j \in [1,N_g]$).

(2) If the point $t_u^{L_i}\left(t_{ux}^{L_i},t_{uy}^{L_i}\right)$ does not belong to any cluster, the circular distance $Dis\left(t_u^{L_i},O_j\right)$ corresponding to the point $t_u^{L_i}$ and each extended circle is calculated by Equation (11), then sort $Dis\left(t_u^{L_i},O_j\right)$ in ascending order, and assign point $t_u^{L_i}$ to the cluster corresponding to the smallest distance,

$$Dis\left(t_u^{L_i},O_j\right) = \left|d^H\left(t_u^{L_i},O_j\right)-R_j\right| \tag{11}$$

where $R_j$ ($R_j \in R$) is the radius the extended circle of cluster $Clu_J$, $j \in [1,N_g]$.

After processing the data of layer $L_i$ according to the above two cases, update the radius and center of the circle according to Equations (7) and (8), and then continue to segment the data of layer $L_{i+1}$ until all data processing is completed, that is, the extraction of individual trees is completed. A diagram of these two cases is shown in Figure 9. In addition, the individual trees of scene 1 obtained by the layer-by-layer radius expansion method is displayed in Figure 10.

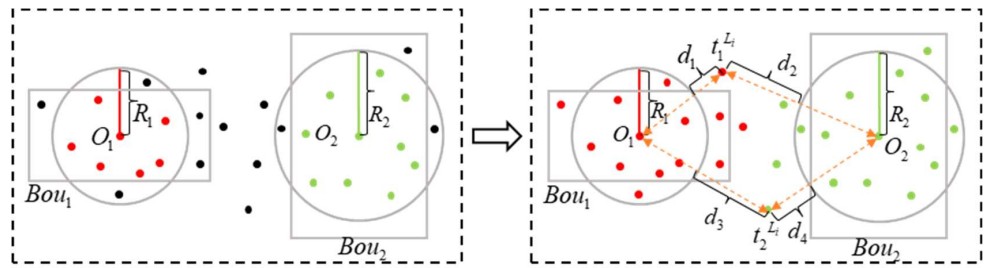

**Figure 9.** Radius expansion method (Points that are located in both the bounding box $Bou_1$ and the extended circle $O_1$ area is marked in red and assigned to the tree where the circle $O_1$ is located, and the points located in the bounding box $Bou_2$ and the extended circle $O_2$ area are marked in green, belongs to the tree where the circle $O_2$ is located. For unclassified black points outside the area, the distances from point $t_1^{L_i}$ to the boundaries of circle $O_1$ and circle $O_2$ are $d_1$ and $d_2$ respectively, and $d_1 < d_2$, so $t_1^{L_i}$ is assigned to the cluster where the circle $O_1$ is located, similarly, compare the distances from point $t_2^{L_i}$ to the boundary of circle $O_1$ and circle $O_2$. Because $d_3 > d_4$ $t_1^{L_i}$ is assigned to the category of circle $O_2$).

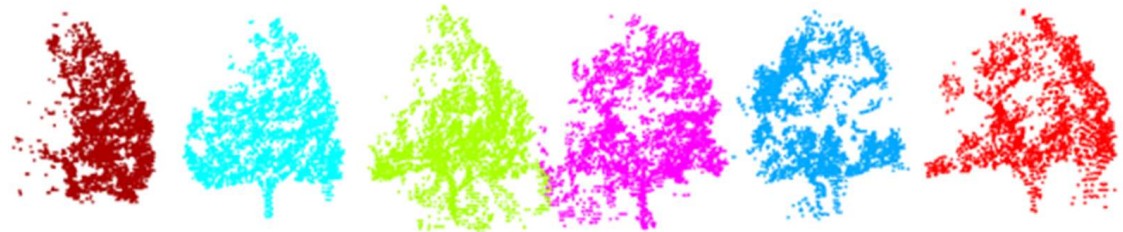

**Figure 10.** Individual trees extraction result of scene 1.

## 3. Results and Discussion

To verify the effectiveness and robustness of our proposed method, experiments are performed on the Oakland 3D Point Cloud dataset. Our method is implemented using C++ and run on a desktop PC with an Intel I5-8500 and NVIDIA GeForce GTX 1660Ti graphics card.

### 3.1. Dataset

The Oakland 3D Point Cloud dataset provided by Munoz et al. [33] is used to verify the effectiveness of the proposed method. The Oakland 3D Point Cloud dataset contains 1.6 million 3D points, consisting of two subsets, part2 and part3, where each scene contains approximately 100,000 3D points. The 3D data from Oakland 3D Point Cloud was acquired using a side-looking SICK LMS lidar MLS system and the dataset was collected near the University of Chicago campus in Oakland, Pennsylvania, and Pittsburgh, Pennsylvania. The dataset is expressed in ASCII format file, and the expression format is {x, y, z, label, confidence}, that is, the three-dimensional space coordinates, label, and confidence of the PCD six information. In addition, a label count file (*. Stats) is provided, which counts the number of points of different categories in each scene. The Oakland 3D Point Cloud dataset roughly classifies 3D point clouds into the following categories: facades, ground, trees, wires, and poles, as shown in Figure 11. This paper simplifies the data categories into trees and non-trees, that is, transforms the semantic segmentation problem into a binary classification problem.

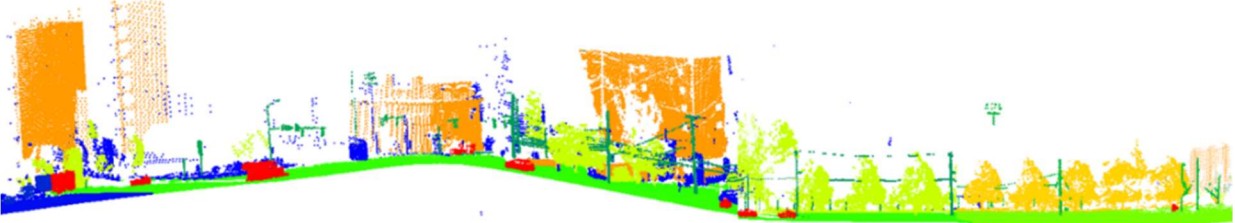

**Figure 11.** Part of Oakland 3D Point Cloud dataset. The dataset roughly classifies 3D point clouds into the following categories with different labels: facades, ground, trees, wires, and poles.

### 3.2. Scene Semantic Segmentation Analysis

We used the Intersection over Union (*IoU*) of each category, the Mean Intersection over Union (*mIoU*) of all categories, and the Overall Accuracy (*OA*) to evaluate the effect of semantic segmentation. *IoU* is the intersection of the network prediction result and the real value compared to their union, *mIoU* is the result of summing and averaging the *IoU* of each category, and *OA* is the ratio of the number of correctly classified samples to the total number of samples. The calculation methods of *IoU*, *mIoU* and *OA* are computed by Equation (12), Equation (13), and Equation (14), respectively.

$$IoU = \frac{TP}{FN + FP + TP} \tag{12}$$

$$mIoU = \frac{1}{k+1} \sum_{i=0}^{k} \frac{TP}{FN + FP + TP} \tag{13}$$

$$OA = \frac{TP + TN}{FN + FP + TP} \tag{14}$$

where *TP* is the actual number of points on the tree, $FP = Nalgo - TP$, *Nalgo* represents the number of tree points detected in the scene, $FN = Nref - TP$, *Nref* and represents the number of tree points marked as true values in the original scene.

Six scenarios are selected as the test set for semantic segmentation of the original PointNet and the multi-feature PointNet network. The original network is a PointNet that

only contains XYZ information, and the method in this paper is a PointNet that contains XYZ information and six local features.

The quantitative evaluation results are displayed in Table 1. As can be seen from Table 1, applying multi-features data to the semantic segmentation network can improve the segmentation results to different degrees in *OA*, *mIoU*, and *IoU* of each category. The OA is above 95%, and the average correct rate reaches 97.8%, which is 5% higher than that before feature fusion, and the *mIoU* is improved by 9.5%. From the *IoU* results of each type of object, after adding local features, the *IoU* of trees and non-trees has been greatly improved. Among them, the *IoU* of trees is significantly improved, which is 13.5% higher than that of the original PointNet network, and the *IoU* of non-tree point clouds is also improved by 5.5%. It can be seen that the local information of the point cloud can enhance the ability of network semantic segmentation effectively.

**Table 1.** Overall Accuracy and Mean *IoU* of six scenes in Oakland 3D Point Cloud dataset.

| The Raw Scene | *OA* (%) | | *mIoU* (%) | | *IoU* (%) Tree | | *IoU* (%) Non-Tree | |
|---|---|---|---|---|---|---|---|---|
| | PointNet | Ours | PointNet | Ours | PointNet | Ours | PointNet | Ours |
| Scene1 | 94.56 | 97.23 | 87.93 | 93.69 | 83.34 | 91.30 | 92.53 | 96.08 |
| Scene2 | 88.17 | 96.15 | 76.19 | 91.05 | 68.25 | 87.35 | 84.13 | 94.76 |
| Scene3 | 96.49 | 97.30 | 92.50 | 94.21 | 90.17 | 92.48 | 94.83 | 95.95 |
| Scene4 | 96.90 | 98.70 | 93.31 | 97.08 | 91.18 | 96.07 | 95.45 | 98.09 |
| Scene5 | 96.53 | 98.81 | 91.22 | 96.81 | 86.94 | 95.18 | 95.49 | 98.44 |
| Scene6 | 87.39 | 98.60 | 68.99 | 94.60 | 52.66 | 90.83 | 85.33 | 98.38 |
| average | 93.34 | 97.80 | 85.02 | 94.57 | 78.76 | 92.20 | 91.29 | 96.95 |

Figure 12 demonstrates the comparison of the semantic segmentation results of the four scenes. The black boxes in Figure 12 indicate the difference between PointNet and our method. Four scene data can be more finely segmented (e.g., wires and utility poles) based on our method. However, with PointNet most of these small objects are wrongly classified as trees. Especially in the scene 4, part of the ground is wrongly divided into trees.

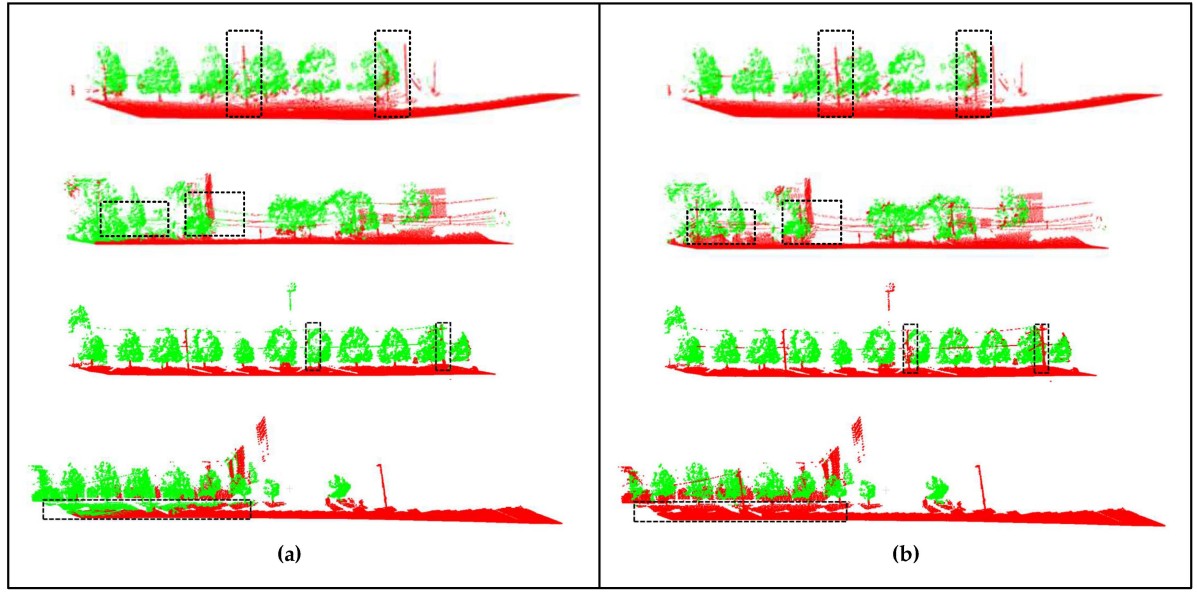

(a)  (b)

**Figure 12.** Semantic segmentation result on four scene data. (**a**) is the result of semantic segmentation, (**b**) is the result of semantic segmentation based on multi-features.

### 3.3. Analysis of Individual Trees Extraction Results

Figures 13 and 14 shows the single tree extraction process diagram of scene 2 and scene 4. Scene 2 contains multiple trees with different shapes and sizes, and there are also cases where tree crowns are connected together. Scene 4 that the PCD in this scene is incomplete and has uneven density. Local maxima and candidate treetop points can be successfully extracted and merged and optimized in two different cases (shown in Figures 13 and 14b–e). The single tree extraction result shown in Figures 13f and 14f is obtained by radius expansion. It can be seen from the result that this method can accurately extract connected individual trees and can also correctly extract trees with obvious crown differences.

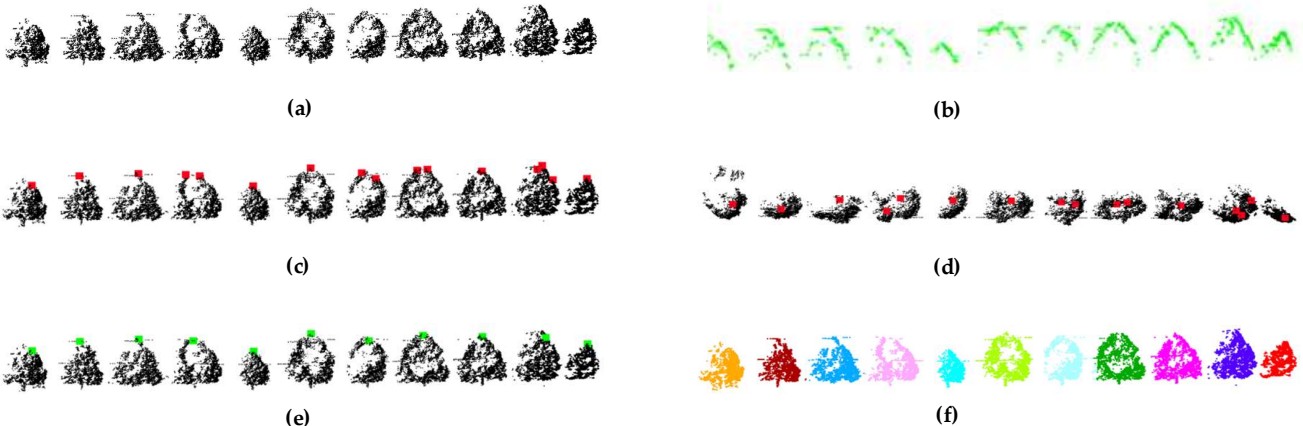

**Figure 13.** Individual trees extraction process of scene 2. (**a**) The PCD of tree, (**b**) local maxima extraction results, (**c**) the side view of the candidate treetop points, (**d**) the front view of the candidate treetop points, (**e**) optimal treetop points, (**f**) individual tree extraction result.

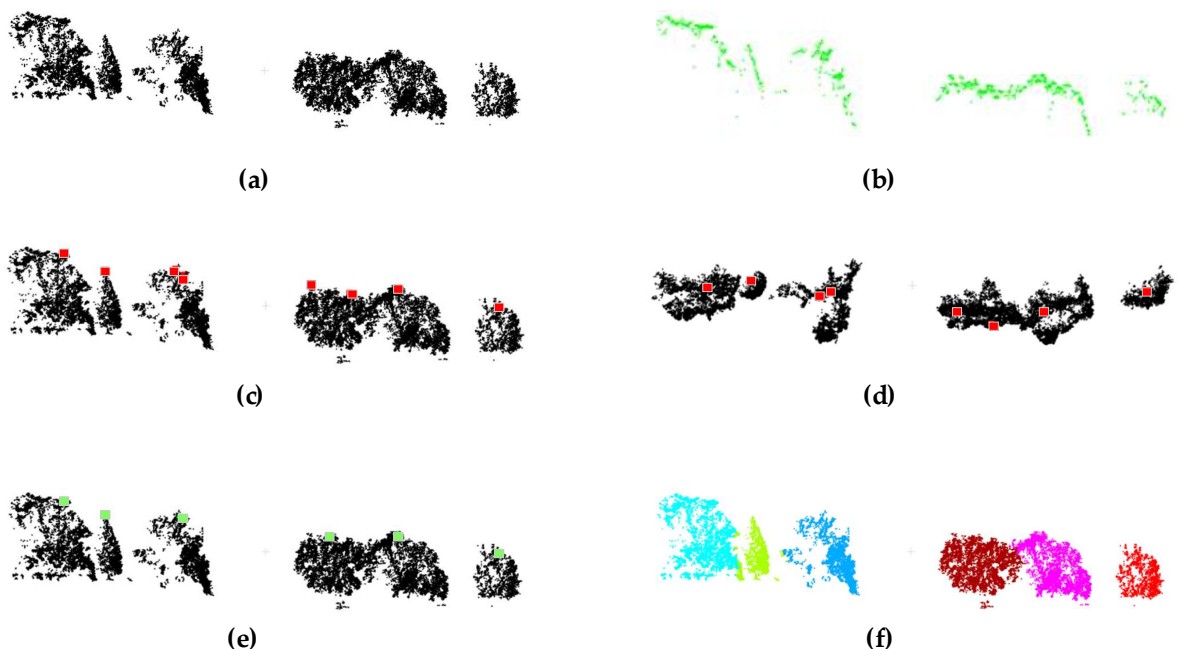

**Figure 14.** Individual trees extraction process of scene 4. (**a**) The PCD of tree, (**b**) local maxima extraction results, (**c**) the side view of the candidate treetop points, (**d**) the front view of the candidate treetop points, (**e**) optimal treetop points, (**f**) individual tree extraction result.

Experimental results demonstrate that the single tree extraction method based on treetop points detection and radius expansion can correctly extract individual trees in outdoor scenes, and the extraction results are not affected by incomplete data and partial tree crown collapse.

### 3.4. Comparative Analysis

Moreover, our proposed method is compared with the voxel-based clustering method [27] and the horizontal slice-based method (3D Forest) [34]. Figure 15 illustrates the experimental results of different methods on five scene datasets. The clustering-based method removes the ground by region growing and then uses the Euclidean clustering algorithm to segment the non-ground points. This method is simple and easy to implement, but due to the existence of various objects in urban outdoor scenes, it is prone to under-segmentation problems when non-tree elements are adjacent to trees or connected to multiple trees. The 3D Forest [34] divides the scene into slices and then divides a single tree according to the number of points in the slice clusters and the distance and angle between the clusters.

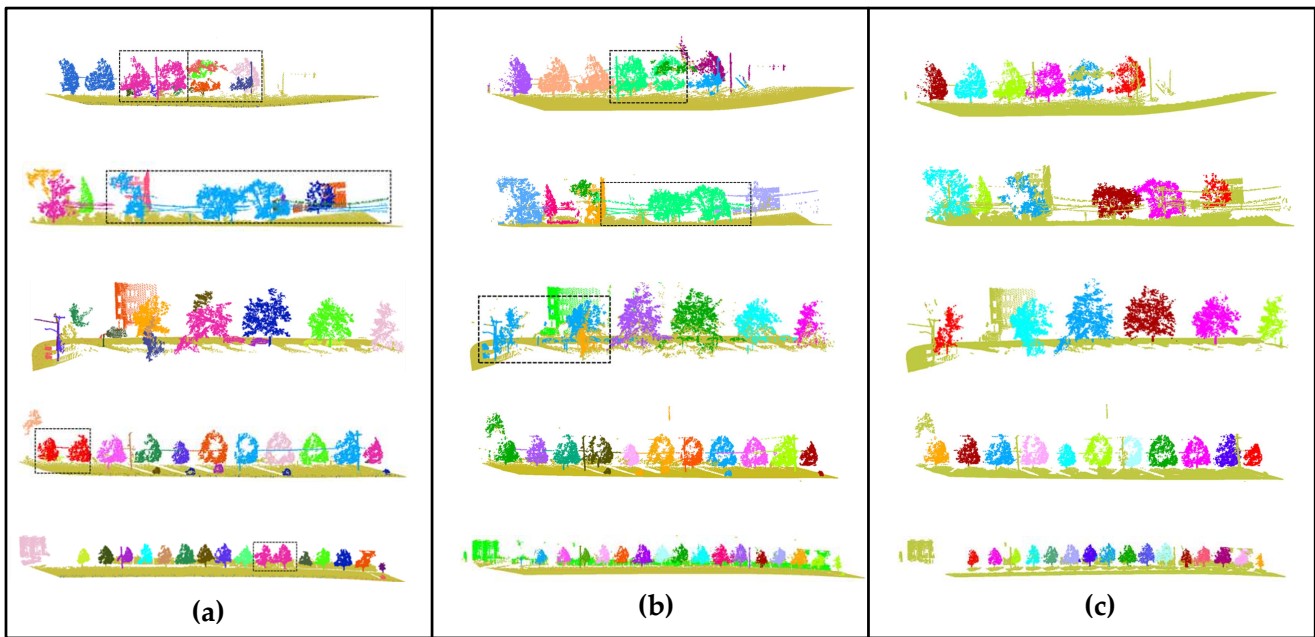

**Figure 15.** Comparison of visual results. (**a**) is the result of the clustering-based method, (**b**) is the result of the 3D Forest, (**c**) is the result of is our method. The black boxes in (**a**,**b**) are misclassification results, and our method could get the result in (**c**).

Compared with these two methods, our proposed method could make finer segmentation and the extraction result is more accurate. Our proposed method can classify trees and non-tree objects in the semantic segmentation stage, and the radius expansion-based method can make full use of the characteristics of trees, and effectively overcome the impact of data missing through top-down hierarchical expansion. Compared with clustering-based method (see in Figure 15a) and 3d Forest (see in Figure 15b), our tree extraction results are closer to real trees, as shown in Figure 15c.

To verify the effectiveness of the proposed algorithm, we analyzed the experimental results quantitatively through six indicators. $TP$(True Positive) represents the number of correctly extracted individual trees, $FN$(False Negative) represents the number of undetected single trees, that is, a single tree and nearby trees are divided into the same tree, $FP$(False Positive) indicates the number of non-trees detected as trees, that is, a point cluster that is not a tree is regarded as a tree. $TP$, $FN$ and $FP$ represent correct segmentation, under-segmentation, and over-segmentation cases respectively. $P$(Precision) is the preci-

sion rate, indicating the proportion of the number of correctly extracted trees to all detected trees, *R* (Recall) is the recall rate, indicating the proportion of the number of correctly extracted trees to the actual trees, *F*(F-score) is a comprehensive index used to evaluate the overall accuracy of tree extraction. The values of *P*, *R* and *F* are calculated according to Equation (15):

$$\begin{cases} P = \frac{TP}{TP+FP} \\ R = \frac{TP}{TP+FN} \\ F = 2 \times \frac{P \times R}{P+R} \end{cases}, \tag{15}$$

The quantitative results of the three methods are listed in Table 2. It can be seen that among the three methods, the accuracy of clustering-based method [26] is the worst. This is because the clustering-based method is prone to under-segmentation when trees are connected with other elements. For example, the three trees connected to electric wires cannot be extracted separately (shown in the second row of Figure 15a). The 3D Forest method [34] is better than the clustering-based method, but there are still over-segmentation and under-segmentation of trees. Comparison experimental results could demonstrate that our proposed method is better than other two methods. For example, for Figure 14a with tree crown overlap, the precision, recall and F-score of the proposed method reach 98.33%, 98.33% and 98.33%, respectively, which are higher than 62.75%, 62.08%, 62.39% of the 3D Forest method that is second only to our method.

**Table 2.** Quantitative comparison results on five scenes.

| The Raw Scene | Method | TP | FP | FN | P | R | F |
|---|---|---|---|---|---|---|---|
| | Clustering method [27] | 2 | 4 | 4 | 0.3333 | 0.3333 | 0.3333 |
| Scene1 | 3D Forest [34] | 1 | 4 | 5 | 0.200 | 0.1667 | 0.1818 |
| | Ours | 6 | 0 | 0 | 1 | 1 | 1 |
| | Clustering method [27] | 2 | 3 | 3 | 0.4000 | 0.2000 | 0.2667 |
| Scene2 | 3D Forest [34] | 2 | 4 | 4 | 0.3333 | 0.3333 | 0.3333 |
| | Ours | 6 | 0 | 0 | 1 | 1 | 1 |
| | Clustering method [27] | 4 | 5 | 5 | 0.4444 | 0.4444 | 0.4444 |
| Scene3 | 3D Forest [34] | 4 | 2 | 2 | 0.6667 | 0.6667 | 0.6667 |
| | Ours | 6 | 0 | 0 | 1 | 1 | 1 |
| | Clustering method [27] | 10 | 2 | 1 | 0.8333 | 0.9091 | 0.8696 |
| Scene4 | 3D Forest [34] | 11 | 0 | 0 | 1 | 1 | 1 |
| | Ours | 11 | 1 | 1 | 0.9167 | 0.9167 | 0.9167 |
| | Clustering method [27] | 15 | 1 | 1 | 0.9375 | 0.9375 | 0.9375 |
| Scene5 | 3D Forest [34] | 15 | 1 | 1 | 0.9375 | 0.9375 | 0.9375 |
| | Ours | 16 | 0 | 0 | 1 | 1 | 1 |
| | Clustering method [27] | - | - | - | 0.5897 | 0.5649 | 0.5703 |
| average | 3D Forest [34] | - | - | - | 0.6275 | 0.6208 | 0.6239 |
| | Ours | - | - | - | 0.9833 | 0.9833 | 0.9833 |

## 4. Conclusions

In this paper, a new method is proposed for individual trees detection from MLS point clouds, which can be used in street tree 3D modeling, street tree monitoring, tree species identifying, and biomass estimation. Our method consists of (1) non-tree points removal and tree detection via multi-feature enhanced PointNet, (2) locating treetop points via filtering the tree point projection plane and optimized treetop points by a distance rule, and (3) after the initial clustering of treetop points and vertical layering of tree points, a top-down layer-by-layer segmentation based on radius expansion to realize the complete individual extraction of trees. The experimental results derived from the Oakland 3D Point Cloud dataset demonstrate that benefiting from the accuracy of scene semantic segmentation and the proposed method can effectively extract the individual

trees. Compared with the other two methods, the proposed method can effectively avoid the influence of artificial roadside pole-like objects and the crown overlaps. Overall, the precision, recall and F-score of instance segmentation on the used datasets are 98.33%, 98.33% and 98.33%, respectively.

In future work, we will improve the robustness of the method to adapt to forests. Additional deep learning can also be explored with goal of improving tree classification accuracy. Meanwhile, the fusion of the orthophoto image and the LiDAR point clouds would provide a better way to greatly improve the efficiency and the accuracy of urban trees detection, especially for the larger scale urban scenes.

**Author Contributions:** Conceptualization, X.N. and Y.H.; Methodology, Y.H.; Software, Y.H.; Validation, X.N., Y.H. and Y.M.; Writing-Original Draft Preparation, X.N., Y.H. and Y.M.; Writing-Review and Editing, X.N., Z.L., H.J. and Y.W. All authors have read and agreed to the published version of the manuscript.

**Funding:** National Natural Science Foundation of China (61871320, 61872291), Shaanxi key Laboratory project (17JS099).

**Data Availability Statement:** The data presented in this study are available on request from the corresponding author. The data are not publicly available due to privacy restrictions.

**Conflicts of Interest:** The authors declare no conflict of interest.

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
