# Peer review of "Semantic Segmentation Guided Coarse-to-Fine Detection of Individual Trees from MLS Point Clouds Based on Treetop Points Extraction and Radius Expansion"

_remotesensing, doi:10.3390/rs14194926_

Round 1

Reviewer 1 Report

The text focuses on the issue of urban planning, distinctly urban trees as elements of outdoor scenes. These components of the built environment can be monitored using mobile laser scanning (MLS) to obtain accurate detection of individual trees from a disordered, discrete and dense MLS. The evaluation is an important basis for subsequent analysis of urban management and planning.
Individual trees cannot be easily extracted from the MLS due to, for example, occlusion by other objects in urban scenes.
Researchers proposed a method for hierarchical detection of individual trees from MLS point cloud data (PCD) based on tree vertex extraction and radius expansion.
The effectiveness of the proposed method was tested and evaluated on five street scenes in point clouds from the Oakland MLS outdoor dataset. Furthermore, the proposed method was compared with two existing individual tree segmentation methods. The evaluation and comparison show that the overall accuracy (precision), repeatability (recall) and F-score of instance segmentation were achieved to be 98.33%, 98.33% and 98.33%, respectively. The results indicate that it can be effective and robust to extract individual trees in various complex environments.
The text of the paper is written based on scientific procedures, and it is supplemented with appropriate pictorial documentation. It proceeds from the theoretical approach to application with a simple example, and demonstrates the properties of the proposed method on real data.
From a formal point of view, I have no major comments.

No plagiarism was found. 

Reviewer 2 Report

A novel method is proposed to detect individual trees from PCD by MLS. The paper is largely well written. I have several comments as follows.  

Line 2 hierarchical used in the title is a bit misleading. multilayer may be more proper. Also deep learning may be included in the title.  

Line 116-132 six features exploited may be clearly described by equations additional to the covariance matrix.   

Line 117 if or not all trees in a scene can be characterized by segments of single-row (think of trees alone a curved road). If not, proper assumptions of the propose method may be given in the paper (discussion?).  

Line 208 format is a chaos here  

Line 433 more description of the changes should be given as I found it a bit hard to discern the improvement.  

Line 522 I consider the last paragraph is meaningless, it may be extended to an informative perspective.

Reviewer 3 Report

The paper proposes a segmentation approach for laser-scanned point clouds detecting individual trees. The method is based on PointNet [19], principal component analysis, and feature-based filtering. The results compare favorably with related methods [17, 24].

Some parts of the paper, in particular the method description, should be re-written in a more focussed manner. Example:

In line 193, the use of PCA (principal component analysis) is proposed to construct a local coordinate system for each tree. However, PCA has been described in  lines 120 ff and eqn (1), already, without mentioning the keyword PCA, at all.

Throughout the paper, there are similar descriptions explaining simple things in a complicated way. It would help a lot, when simplifying these explanations.

Example: in eqn (3) n is used as centroid and v_2=(a,b,c) is a normal vector (which would be better denoted with n). The square root is simply ||v_2||^2.

Further Issues:

Title: "Individual Tree" -> "Individual Trees"

within the paper, either use "an individual tree" (in case there is only one) or "individual trees" (when there are multiple).

line 144: Please, explain the elements of the 9D-vector.

148 ff: Justify the dimensions of matrices used.

193 PCA is mentioned here, but it has been explained in lines 120 ff and eqn (1), already

204 ff: a lot of symbols that do not belong here appear on top of text and formulae.

211: Does nt_l denote a vector between n and t_l? Then, use tne norm ||.|| rather than abs.

373 Results and discuss(ion)

most figures need to be much greater to show something significant

online-sources like [22] may not be listed as literature. Better use footnotes.

There exist many more references on the topic, for example see:

Aaron M. Sparks, Mark V. Corrao, Alistair M. S. Smith:

Cross-Comparison of Individual Tree Detection Methods Using Low and High Pulse Density Airborne Laser Scanning Data. Remote. Sens. 14(14): 3480 (2022)

Gábor Brolly, Géza Király, Matti Lehtomäki, Xinlian Liang:

Voxel-Based Automatic Tree Detection and Parameter Retrieval from Terrestrial Laser Scans for Plot-Wise Forest Inventory. Remote. Sens. 13(4): 542 (2021)

Lukasz Kolendo, Marcin Kozniewski, Marek Ksepko, Szymon Chmur, Bozydar Neroj:

Parameterization of the Individual Tree Detection Method Using Large Dataset from Ground Sample Plots and Airborne Laser Scanning for Stands Inventory in Coniferous Forest. Remote. Sens. 13(14): 2753 (2021)

Karel Kuzelka, Martin Slavík, Peter Surový:

Very High Density Point Clouds from UAV Laser Scanning for Automatic Tree Stem Detection and Direct Diameter Measurement. Remote. Sens. 12(8): 1236 (2020)

Lloyd Windrim, Mitch Bryson:

Detection, Segmentation, and Model Fitting of Individual Tree Stems from Airborne Laser Scanning of Forests Using Deep Learning. Remote. Sens. 12(9): 1469 (2020)

Christoph Gollob, Tim Ritter, Clemens Wassermann, Arne Nothdurft:

Influence of Scanner Position and Plot Size on the Accuracy of Tree Detection and Diameter Estimation Using Terrestrial Laser Scanning on Forest Inventory Plots. Remote. Sens. 11(13): 1602 (2019)

Wuming Zhang, Peng Wan, Tiejun Wang, Shangshu Cai, Yiming Chen, Xiuliang Jin, Guangjian Yan:

A Novel Approach for the Detection of Standing Tree Stems from Plot-Level Terrestrial Laser Scanning Data. Remote. Sens. 11(2): 211 (2019)

Carlos Cabo, Celestino Ordóñez, Carlos A. López-Sánchez, Julia Armesto:

Automatic dendrometry: Tree detection, tree height and diameter estimation using terrestrial laser scanning. Int. J. Appl. Earth Obs. Geoinformation 69: 164-174 (2018)

Ivar Oveland, Marius Hauglin, Francesca Giannetti, Narve Schipper Kjørsvik, Terje Gobakken:

Comparing Three Different Ground Based Laser Scanning Methods for Tree Stem Detection. Remote. Sens. 10(4): 538 (2018)

Reviewer 4 Report

Paper is well written and methodological approach is straightforward and easy to understand. Context and challenges are well described. I would like to highlight the nice quality of figures, but all figures should be enlarged to ease the reading.

Some remarks:

L126: comma at the beginning of the line to be removed.

L204 to 211: strange artefacts on the pdf?

L373: discussION ?

L456: blue color with white background is not the best choice

L457: Figure 14 in bold

Do the author see complementarity with aerial EO data, such as LiDAR point clouds or orthophotos? This could be discussed in the perspectives to provide insights on which approach to choose to delineate urban trees at larger scale.
